# Synthesis Antifreezing and Antidehydration Organohydrogels: One-Step In-Situ Gelling versus Two-Step Solvent Displacement

**DOI:** 10.3390/polym12112670

**Published:** 2020-11-12

**Authors:** Chun Li, Xiaobo Deng, Xiaohu Zhou

**Affiliations:** College of Chemistry and Environmental Engineering, Shenzhen University, Shenzhen 518060, China; lichun2018@email.szu.edu.cn (C.L.); 1900221020@email.szu.edu.cn (X.D.)

**Keywords:** organohydrogel, antifreezing, antidehydration, in-situ gelling, solvent displacement

## Abstract

Organohydrogels with distinct antifreezing and antidehydration properties have aroused great interest among researchers, and various organohydrogels and organohydrogel-based applications have emerged recently. There are two popular synthesis strategies to prepare these antifreezing and antidehydration organohydrogels: the in-situ gelling and the solvent displacement strategies. Although both strategies have been widely applied, there is a lack of comparative study of these two strategies. In this work, to elucidate the comparative advantages of the two synthesis strategies, we studied and compared the mechanical and environmental tolerant properties of the organohydrogels synthesized from both strategies. The glycerol-based and ethylene glycol-based chemical polyacrylamide (PAAm) organohydrogel and the glycerol-based physical gelatin organohydrogel were synthesized and studied. Through the comparative study, we have found that the organohydrogels from different strategies with the same dispersion medium showed similar antifreezing and antidehydration properties but different mechanical properties. The mechanical properties of these organohydrogels are influenced by two opposite factors for each strategy: the enhanced physical interactions induced strengthening and solvent effect or swelling induced weakening. We hope this study may provide a better understanding of the synthesis strategies of organohydrogels and provide a valuable guide to choose the suitable synthesis strategy for each application.

## 1. Introduction

Hydrogel with unique biocompatible soft and wet properties has been widely applied in various applications [1], such as tissue engineering, drug delivery, sensors and actuators, and so on [1,2,3,4]. However, two major drawbacks limit the applications of the conventional hydrogels: the poor mechanical properties and the weak environmental tolerant properties (antifreezing and antidehydration) [5,6,7,8]. To solve the problem of poor mechanical properties, in the last two decades, various strategies have been developed to fabricate tough and strong hydrogels with enhanced mechanical performances [7,9,10,11]. However, the problem of weak environmental tolerance has been rarely studied [8]. Owing to their aqueous medium in the matrixes, hydrogels would freeze below the freezing point of water and dry by dehydration during long-term storage in the arid or hot environment. The frozen or dried hydrogels would become rigid and brittle and lose their desired properties, such as toughness, strength, stretchability, self-healing properties, conductivity, responsibility, and so on [8].

Recently, inspired by the biological organism to adapting the extreme environment, scientists have reported various synthesis strategies to prepare the antifreezing and antidehydration hydrogels [8,12,13,14,15,16,17,18]. These environment-adaptable antifreezing and antidehydration gels are fabricated by incorporating either the high concentration of salts or organic–water binary solvent in the gel matrix [12,13,14,15,16,17,18]. The general mechanism to obtain the environmental tolerant properties is to form abundant hydrogen bonds between the incorporated salt ions or solvent molecules with water molecules, which would enhance the water retention capacity from evaporation and disturb the hydrogen-bonded network structure of water to prevent ice formation [8,19,20,21].

Owing to the distinct antifreezing and antidehydration properties, the organohydrogel has aroused great interest among researchers since it was introduced [12,13,14,15,22]. Various organohydrogels and organohydrogel-based applications have emerged recently, and the glycerol [14,15,22,23,24,25,26], ethylene glycol (EG) [13,15,27,28,29,30,31], DMSO [32,33], and sorbitol [15,34] are widely used. There are two popular strategies to fabricate organohydrogels. The first involves the one-step in-situ gelling in an organic–water binary solvent [13,14,22]. Liu and coworkers first introduced the antifreezing conductive organohydrogels by utilizing the EG-water binary solvent as the dispersion medium, which endowed the temperature tolerance range from −55.0 to 44.6 °C [13]. At the same time, Lu and coworkers synthesized the adhesive and conductive organohydrogel with long-lasting liquid retention at the temperature range from −20 to 60 °C by in situ polymerization in the glycerol–water binary solvent [14]. Since then, various organohydrogels were prepared by in-situ gelling in the organic–water binary solvent [25,26,29,30,31,32,33,35,36,37,38,39]. The second synthesis strategy is to soak the preformed hydrogel in the organic or organic–water binary solvent for solvent displacement [15]. Zhou and coworkers first prepared a serial of antifreezing and antidehydration tough organohydrogels by soaking the preformed hydrogels in the organic solvent, such as glycerol, glycol, sorbitol, or a mixture of these. After solvent displacement, the obtained organohydrogels exhibited low temperature tolerance down to −70 °C [15]. Since then, due to its universality, solvent displacement became another popular strategy to synthesize the environment-adaptable organohydrogels [23,24,27,28,34,40,41,42,43,44,45,46].

Although both strategies have been widely applied to fabricate the functional organohydrogels, there is a lack of a comparative study between these two strategies to guide the strategy selection. Both strategies have their advantages and disadvantages. For instance, the one-step in-situ gelling in a binary solvent is more facile and convenient, but the solvent effect could significantly hinder or even completely block the gelling process, which makes this strategy not universal [14,27]. In contrast, the solvent displacement strategy should be a universal method due to the unique liquid transport properties of the hydrogels [15]. However, the solvent displacement is a two-step method that requires the preformed hydrogel and the additional soaking step. Furthermore, the soaking induced swelling would change the volume and shape of the gel and then weaken the matrix.

To clarify the comparative advantages of the two synthesis strategies of organohydrogels, in this work, we studied and compared the mechanical and environmental tolerant properties of the organohydrogels synthesized from these two strategies. To systemically investigate and compare these two synthesis strategies, two widely used hydrogel systems are chosen for this study, the chemically crosslinked polyacrylamide (PAAm) hydrogel, and the physically crosslinked gelatin hydrogel. Additionally, the cryoprotectants of glycerol and ethylene glycol are applied as the organic solvent in the organic–water binary solvent. Through the comparative study, we have found that the organohydrogels from different strategies with the same dispersion binary solvent had a similar environmental tolerant capability but different mechanical performances. The mechanical properties of these organohydrogels are influenced by two opposite factors for each synthesis strategy, the enhanced physical interactions induced strengthening and solvent effect or swelling induced weakening. The combined effect determines the mechanical properties of the as-prepared organohydrogels. We hope this study may provide a better understanding of the synthesis strategies of organohydrogels and provide a valuable guide to choose the suitable synthesis strategy for each application.

## 2. Materials and Methods

### 2.1. Materials

Ammonium persulfate (APS, 99.9%), gelatin (~250 g Bloom), and ethylene glycol (AR, 98%) were purchased from Macklin (Shanghai, China). Acrylamide (AAm, 99%), *N*,*N*′-Methylenebisacrylamide (MBAA), and glycerol were purchased from Sigma-Aldrich (St. Louis, MO, USA). Deionized water (DI water) was used in all experiments.

### 2.2. Preparation of Polyacrylamide Hydrogel (PAAm Hydrogel)

A total of 7.2 g AAm monomer was added into 30 mL DI water. Further, 360 mg APS as the photothermal initiator and 21.6 mg MBAA as the crosslinking agent were added to the former solution. Subsequently, the resulting solution was poured about 3.5 g into a rectangular mold measuring 23 × 43 × 19 mm^3^ and cured with UV light at a wavelength of 365 nm (25 W, ZF-5, Shanghaijiapeng, Shanghai, China) at 40 °C for 1 h.

### 2.3. Preparation of Gelatin Hydrogel

A total of 15 g gelatin was added into 85 mL DI water with stirring and 70 °C water bath heating for 2 h. Subsequently, the resulting solution was poured about 3.5 g into a rectangular mold measuring 23 × 43 × 19 mm^3^ and gelled by cooling down to room temperature.

### 2.4. Preparation of Organohydrogel by Solvent Displacement

The PAAm hydrogel and gelatin hydrogel were soaked into the organic–water binary solvent, with the volume ratio of the organic solvent to the water as 1:4, 1:2, 1:1, 2:1, and 4:1 for 3 days.

The PAAm hydrogel and gelatin hydrogel were soaked into the organic solvent (glycerol or EG) for 3 days and the organic solvent was replaced with fresh solvent per 12 h to prepare the organogels. 

### 2.5. Preparation of PAAm Organohydrogel by In-Situ Gelling

In total, 7.2 g AAm monomer was added into DI water. Subsequently, 360 mg APS and 21.6 mg MBAA were added to the former solution. Subsequently, the organic solvent was added to the above solution. The contents of all materials in the gel preparation solution are shown in Table 1. The resulting solution was poured about 3.5 g into a rectangular mold measuring 23 × 43 × 19 mm^3^ and cured with UV light at a wavelength of 365 nm at 40 °C for 1 h. The cured mixture was left in a humid box overnight to stabilize the reactions. 

### 2.6. Preparation of Gelatin Organohydrogel by In-Situ Gelling

Gelatin was added into DI water with stirring and 70 °C water bath heating for 2 h. Subsequently, the glycerol was added to the gelatin solution for 5 min. The contents of all materials in the gel preparation solution are shown in Table 2. The resulting solution was poured about 3.5 g into a rectangular mold measuring 23 × 43 × 19 mm^3^ and gelled by cooling down to room temperature.

### 2.7. Mechanical Properties of Gels

For stress-strain tensile tests, the hydrogel samples were stretched until the fracture with the velocity of 20 mm min^−1^. The test of mechanical properties all used the tensile testing machine (CMT6103, SANS, MTS, Eden Prairie, MN, USA). The elastic modulus was determined from the slope stress-strain curve at the strain between 0 to 10%. The toughness was calculated by integrating the stress-strain curve.

### 2.8. Antiheating and Antidehydration Investigation

The weight of the organohydrogels was measured (w_0_). Then, these gels were placed in the 60 °C hot oven for heating or in a humid box at room temperature for long-term storage. The humid box was kept at 50% humidity. The organohydrogels were stored in the humid box for 15 days. Subsequently, the weight of the antidehydration gels (w) was measured and calculated the remaining weight ratio (w/w_0_) was calculated.

### 2.9. Equilibrium Swelling Ratio

During the solvent displacement, the hydrogels would swell or shrink. The weight of the original hydrogels was measured (*W*_0_). After solvent displacement in the organic–water binary solvents, the weight of the organohydrogels was measured (*W*). The swelling ratio by weight was defined as *W/W*_0_. 

The swelling ratio by volume was calculated from the weights, since it was difficult to precisely measure the volume of the swollen gels. The original volume of the hydrogels was defined as the *V*_0_ = (*W*_0_ − *W_dry_)/d_water_*, where *W_dry_* was the weight of the hydrogels in dry state and *d_water_* was the density of water. The volume of the gels after solvent displacement was defined as the *V = (W − W_dry_)/d_solvent_* where *d_solvent_* was the density of the organic–water binary solvents (Appendix A). Therefore, the swelling ratio by volume was defined as *V/V*_0_.

### 2.10. Differemtial Scanning Calorimetry (DSC) Investigation

Glycerol-based PAAm organohydrogels were characterized using a differential scanning calorimeter (DSC 200 F3, Netzsch, Selb, Germany). The cooling cycle was performed from 50 to −120 °C at a rate of 5 °C min^−1^. Subsequently, the heating cycle was performed from −120 to 50 °C at the rate of 5 °C min^−1^.

### 2.11. Rheological Investigation

Rheological investigation of the glycerol-based PAAm organohydrogels was conducted on the rheometer (MCR 302, Anton Paar, Graz, Austria) with 25 mm flat parallel plates. The frequency and strain amplitude sweep tests were performed at T = 25 °C. The frequency dependence of storage modulus (*G’*) and loss modulus (*G’’*) were performed at 0.5% of shear strain (g = 0.5%) over a frequency range from 0.1 to 100 rad s^−1^.The shear strain dependence of *G’* and *G’’* were performed at w = 10 rad s^−1^ over a shear strain from 0.01 to 100%. The temperature ramp with a cooling and heating rate of 5 °C min^−1^ at w = 10 rad s^−1^ and g = 0.5% was conducted over the temperature range from −10 to 80 °C.

## 3. Results and Discussions

### 3.1. Synthesize Glycerol-Based PAAm Organohydrogels from Both Strategies

The scheme in Figure 1 illustrates the two synthesis strategies to synthesize PAAm organohydrogels (Appendix A). The in-situ gelling strategy is a one-step method. All reagents were dissolved in the organic–water binary solvent and the organohydrogel was formed directly from this precursor binary solution (Figure 1a). In contrast, the solvent displacement strategy is a two-step method. First, the PAAm hydrogel was formed from the precursor aqueous solution which contained all the reagents. Then, the preformed hydrogel was immersed in the organic (glycerol or EG) or organic–water binary solvent for solvent displacement to obtain the PAAm organohydrogel (Figure 1b). To systemically study and compare these two synthesis strategies, the binary solvents with different volume fractions of organic solvent were used (from 0 to 100%). 

During the synthesis with the in-situ gelling strategy, we have found that the glycerol content would significantly hinder the in-situ polymerization of the PAAm in the binary solvent. As the glycerol content increased, more crosslinker and initiator were required to form the organohydrogel matrix (Appendix A). This solvent effect of the binary solvent should be the combined effect of the lower solubility of the monomers, higher viscosity of the solution, and the chemical activity of the abundant functional groups which will react and quench the radicals during polymerization. However, for the solvent displacement, there was no such problem, because the gel matrix was preformed in an aqueous medium. These results confirmed that the solvent displacement was a more universal strategy to transfer the hydrogels to the organohydrogels. Therefore, to facilitate the direct comparison of the mechanical and environmental tolerant properties between the organohydrogels from different strategies, the recipe of the PAAm gel system with 5 wt % of acrylamide with 0.3 wt % MBAA and 5 wt % APS was used (Materials and Methods, Appendix A). Due to the solvent effect of the in-situ gelling strategy, the PAAm organogel with 100% glycerol could not be obtained with the in-situ gelling strategy, and the PAAm organohydrogels with the glycerol content from 20% to 80% were prepared. In contrast, the PAAm organohydrogels with the glycerol content from 20% to 80%, and the PAAm organogel were successfully prepared with the solvent displacement strategy.

### 3.2. Mechanical Properties of the Glycerol-Based PAAm Organohydrogels

The mechanical properties of these PAAm organohydrogels were determined (Figure 2). The mechanical properties of the PAAm organohydrogel synthesized from in-situ gelling strategy might be influenced by two opposite factors. First, as the glycerol content was introduced, more hydrogen bonds should form between the solvent molecules and PAAm chains, which would enhance the mechanical properties of the PAAm organohydrogels [14]. However, on the other hand, the glycerol-water binary solvent would generate the solvent effect, which would hinder the polymerization of polymers and weakens the mechanical properties. As shown in Figure 2, compared to the PAAm hydrogel, the PAAm organohydrogel became softer and more stretchable. As the glycerol content increased, the tensile strain of the PAAm organohydrogel monotonically increased from 1.73 to 4.58, while the tensile stress and elastic modulus monotonically decreased from 38.42 kPa and 57.21 kPa to 16.24 kPa and 9.38 kPa, respectively. These results indicated that the solvent effect dominated the mechanical performance of the PAAm organohydrogels and the higher glycerol content could generate a larger solvent effect. In contrast, the hydrogen bonding-induced strengthening effect was relatively weaker. Interestingly, the toughness of the PAAm organohydrogels with different glycerol content were all similar to that of the PAAm hydrogel, which might come from the balancing between the two opposite effects.

For the glycerol-based PAAm organohydrogels from solvent displacement, there were also two opposite factors that might significantly influence their mechanical properties. First, glycerol molecules would form more hydrogen bonds with polymers, which were inclined to strengthen the matrix. However, the soaking-induced swelling during the solvent displacement would weaken the mechanical properties. The swelling ratios of the PAAm gels decreased as the glycerol content increased (Appendix A). Due to the large swelling ratio of 391% by volume, the tensile strain, tensile stress, modulus, and toughness decreased from 1.73, 38.42 kPa, 57.21 kPa, and 43.30 kJ m^−3^ of the PAAm hydrogel to 0.39, 7.63 kPa, 26.69 kPa, and 1.75 kJ m^−3^ of the PAAm organohydrogel with 20% glycerol, respectively. Furthermore, the PAAm organohydrogel with 20% glycerol showed the similar mechanical performance with the PAAm hydrogel at the equilibrium swollen state in water (Appendix A), which indicated that the swelling-induced weakening effect dominated the mechanical performance. Then, as the glycerol content increased, the mechanical properties gradually recovered due to the decreased swelling ratio and increased hydrogen bonds. When the glycerol content was 80%, the swelling ratio was about 1.12 in volume, and the mechanical properties of the PAAm organohydrogel recovered back with the tensile strain of 2.17, the tensile stress of 45.78 kPa, modulus of 65.54 kPa, and toughness of 67.41 kJ m^−3^. The enhanced mechanical performance of the PAAm organohydrogel with 80% glycerol to the PAAm hydrogel could be estimated as the contribution of the glycerol-induced strengthening. When the glycerol content approached 100%, the gel matrix deswelled and the PAAm organohydrogel almost became the PAAm organogel, which exhibited the best mechanical performance due to the volume shrinkage and abundant hydrogen bonds.

Rheological behaviors of these glycerol-based PAAm organohydrogels were also investigated with the frequency sweep test, strain amplitude sweep test, and temperature ramp test (Materials and Methods). In the whole frequency range from 0.1 to 100 rad s^−1^, the storage modulus (*G’*) was larger than the loss modulus (*G”*) for each sample, which indicated the solid-like behavior and elastic nature of gels. Interestingly, for the organohydrogels from in-situ gelling, the gel with more glycerol content was more viscous (higher loss factor) than gel with less glycerol, which was consistent with that the solvent effect increased as the glycerol content increased. The stronger solvent effect could hinder the polymerization more. However, for the organohydrogels from solvent displacement, the loss factors showed a different trend, and the gel with the medium glycerol content (33%) showed the most viscous behavior (Appendix A). The viscoelastic response of these organohydrogels was dependent on shear strain (Appendix A). The region where complex modulus (*G**) was paralleled with strain was the linear viscoelastic region, and the limited strain of the linear viscoelastic region was defined as the critical strain (γ_c_). All organohydrogels from in-situ gelling had a similar linear viscoelastic region up to 10% of strain, which was similar to the PAAm hydrogels. However, the linear viscoelastic region of the organohydrogels from solvent displacement was dependent on the glycerol content of the gels. The organohydrogel with 33% glycerol had the minimum critical strain of about 0.8%, which also showed the highest loss factor in the frequency sweep test (Appendix A). The viscoelastic response of these organohydrogels was also dependent on temperature (Appendix A). In the whole temperature range from −10 to 80 °C, *G’* was larger than *G”* for each sample, indicating the solid-like behavior. For the organohydrogels from in-situ gelling, the loss factor decreased as the temperature increased, which was similar to the PAAm hydrogel. Interestingly, for the organohydrogels from solvent displacement, the gels with moderate glycerol content (from 33% to 67%), the loss factors could almost keep constant for a wide temperature range (Appendix A).

### 3.3. Antifreezing Properties of the Glycerol-Based PAAm Organohydrogels

Glycerol is a widely used antifreezing agent, and the freezing point of the glycerol–water mixture solution could be reduced to −40 °C [14,20]. To investigate the antifreezing properties, these PAAm gels were placed at −40 °C for 24 h. The PAAm organohydrogels from both strategies showed similar results (Figure 3a and Appendix A). The PAAm hydrogel was completely frozen and became opaque and brittle, and the organohydrogels with less glycerol content also became opaque. However, when the glycerol content in the gel matrix was more than 50%, there were no observable icing phenomena and the organohydrogels could maintain their flexibility and transparency (Figure 3c). Interestingly, although the organohydrogels with less glycerol (20% and 33%) were frozen and lost their transparency, they could still maintain their flexibility (Figure 3b), which indicated that these organohydrogels were only partially frozen and the glycerol in gels prevent the aggregation of the ice crystals in gels. Then, we characterized these gels with the dynamic scanning calorimetry (DSC) from −120 to 50 °C to obtain more accurate phase diagrams. The PAAm organohydrogels from both strategies also showed similar results (Appendix A). PAAm hydrogel showed a sharp peak at −23.16 °C, which should be contributed to the icing of water in the hydrogel. As the glycerol content increased from 0 to 33%, the hydrogels became organohydrogels and the peak shifted to a lower temperature and became broader and smaller, which confirmed that the organohydrogels with less glycerol (less than 50%) were partially frozen (Figure 3a,b). As the glycerol content continuously increased, there was no observable peak for the organohydrogels, which confirmed that there was no observable icing in these organohydrogels. 

### 3.4. Antiheating and Antidehydration of the Glycerol-Based PAAm Organohydrogels

Generally, the relative nonvolatile glycerol could also significantly reduce the saturated vapor pressure by forming a variety of molecular clusters with water molecules, which prevent the evaporation of the water in the hot or arid environment. Therefore, we then investigated the antiheating and antidehydration properties of these organohydrogels by placing them in the 60 °C oven for 24 h or at room temperature with low humidity (25 °C and 50% humidity) for 15 days. Unlike the water-based PAAm hydrogels which completely dried and became brittle, the organohydrogels after heating or long-term storage still maintained moisture and flexibility (Figure 4 and Figure 5). As the glycerol content increased in the gel matrix, less weight was lost. Interestingly, it seemed that when the organohydrogels with the glycerol content less than 50%, organohydrogels from solvent displacement showed larger weight loss than the gels from in-situ gelling by heating (Figure 4a), while the organohydrogels from in-situ gelling could lose more weight by long-term storing (Figure 4b). This “counterintuitive” phenomenon came from two reasons. First, the organohydrogels from solvent displacement contained much more binary solvent than the gels from in-situ gelling due to swelling. Second, compared to the long-term storing at room temperature with low humidity, heating in the hot air oven was a more rigorous treatment (60 °C and 10% humidity), and most of the water in the organohydrogels was lost, and these organohydrogels almost became organogels. Therefore, when the water in organohydrogels was almost lost, the organohydrogels with more binary solvent could have a large weight loss (Figure 4a). 

Since we knew the amount of each component in the organohydrogels, if we assumed that the polymer and glycerol in the matrix would remain and only water would lose during the heating and long-term storage, we could estimate the proportion of the dispersion medium in the remaining gels (Figure 4c,d). These results showed that the organohydrogels with the same initial dispersion medium from different strategies almost had the same glycerol content in the remaining gels after heating or long-term storing, except for the organohydrogels with low glycerol content after heating. When the glycerol content was less than 50%, the organohydrogels from in-situ gelling had a slightly better water retention capability (Figure 4c,d), which might come from the more uniform distribution of the glycerol-water binary solvent. The binary solvent was more uniform, and then more water molecules could form clusters with glycerol molecules, which facilitated the water retention. However, when the glycerol content was larger than 50%, there were enough glycerol molecules to form clusters with water molecules and the distribution effect could be neglected. These results also confirmed that heating treatment could evaporate more water than long-term storage (Figure 4c,d). 

The mechanical properties of the gels after heating at 60 °C oven for 24 h were also determined (Figure 5). For the organohydrogels from in-situ gelling, as the glycerol content decreased, more water evaporated. Additionally, the higher glycerol content could generate a larger solvent effect to weaken the organohydrogels (Figure 2). Therefore, due to the combined effects of the larger volume shrinkage and less solvent-effect-induced weakening, the organohydrogels with less glycerol showed better mechanical properties. The PAAm organohydrogel with 20% glycerol after heating exhibited the best mechanical performance with a tensile strain of 12.32, the tensile stress of 276.54 kPa, the elastic modulus of 207.25 kPa, and toughness of 2309.98 kJ m^−3^ (Figure 5). However, the organohydrogels from solvent displacement showed a different trend, and the mechanical properties first decreased and then increased as the glycerol content increased (Figure 5). Although these organohydrogels came from the same PAAm hydrogels, the swelling ratios and dispersion mediums were different after solvent displacement. When the organohydrogels were heated and lost most of the water, the remaining weights were different (Appendix A). Since the glycerol contents of these gels after heating were similar (Figure 4c), the larger remaining weight meant larger volume and poorer mechanical properties. Therefore, the trend of mechanical properties was concave up as glycerol content increased, which was consistent with the remaining weight of these gels (Appendix A).

### 3.5. Mechanical Properties of the EG-Based PAAm Organohydrogels

To further investigate these two synthesis strategies, we also utilized the EG as the organic solvent to synthesize PAAm organohydrogels [13], and the mechanical properties were determined (Figure 6). As mentioned previously, the mechanical performance of the EG-based organohydrogels from each synthesis strategy might be also influenced by two opposite factors: the hydrogen-bonding-induced strengthening and solvent effect for in-situ gelling or swelling for solvent-displacement-induced weakening. Similar to the glycerol-based organohydrogels from in-situ gelling, the EG-based organohydrogels from in-situ gelling also became softer and more stretchable as the EG content increased. The tensile strain monotonically increased from 1.68 to 5.68, while the tensile strength and modulus monotonically decreased from 30.09 kPa and 58.89 kPa to 9.81 kPa and 4.54 kPa, respectively. These results confirmed that the solvent effect dominated the mechanical performance. For the organohydrogels from solvent displacement, since the swelling ratios with different EG-water binary solvents were almost similar (Appendix A), the mechanical properties of these organohydrogels were also almost similar, except for the EG-based organogel which had a smaller swelling ratio of 200% by volume (Figure 6). These results confirmed that the swelling-induced weakening also dominated the mechanical performance and the strengthening effect of the hydrogen bonding between EG molecules and polymers could be neglected. 

### 3.6. Mechanical Properties of the Glycerol-Based Gelatin Organohydrogels

To further expand the comparison of these two synthesis strategies, besides the chemically crosslinked PAAm gel, we also used the physically crosslinked gelatin gel to prepare gelatin organohydrogels (Figure 7). Gelatin is a mixture of peptides and proteins produced from collagen, which is the main structural protein in various connective tissues in the animal and human bodies. Gelatin can form weak and brittle gels by forming triple-helices and weak physical crosslinked networks. With the great biocompatibility, it is greatly interesting to fabricate tough gelatin gels for biomedical applications [1,23,46]. Here, we fabricated the glycerol-based gelatin organohydrogels with the two synthesis strategies, and the mechanical properties were also determined (Figure 7). Due to the low solubility of the gelatin in glycerol, the gelatin organohydrogels with high glycerol content (more than 50%) could not be formed by in-situ gelling. The glycerol significantly strengthened the gelatin organohydrogels, and the tensile strain, tensile stress, elastic modulus, and toughness increased from 0.31, 5.15 kPa, 29.00 kPa, and 1.21 kJ m^−3^ for gelatin hydrogel to 0.59, 6.71 kPa, 60.73 kPa, and 3.71 kJ m^−3^ for gelatin organohydrogel with 20% glycerol, respectively. Interestingly, as the glycerol content increased from 20% to 50%, the mechanical properties of the gels did not further significantly increase, which might come from the canceling out of the two opposite factors, glycerol-induced strengthening [23] and solvent-effect-induced weakening. In contrast, due to the large swelling ratios of the gelatin hydrogels in the glycerol–water binary solvents (Appendix A), the gelatin organohydrogels with low glycerol content (less than 50%) were too weak to conduct the tensile stress testing (Figure 7). As the glycerol content increased, the mechanical properties of the gelatin gels increased, which came from the combined effects of the decreased swelling ratio and glycerol induced strengthening. With shrinkage to about half of the original volume, the gelatin organogel exhibited extremely strong and tough mechanical performance with the tensile strain of 1.81, the tensile stress of 1044.33 kPa, the elastic modulus of 1455.00 kPa, and toughness of 1007.57 kJ m^−3^. 

## 4. Conclusions

In this work, we have investigated the two popular synthesis strategies for preparing antifreezing and antidehydration organohydrogels, the one-step in-situ gelling and two-step solvent displacement strategies. The chemically crosslinked PAAm organohydrogels with glycerol or EG as the organic solvent and the physically crosslinked gelatin organohydrogels with glycerol were synthesized from both synthesis strategies. The properties of these gels were systematically studied and compared. Through the comparative study of these two synthesis strategies, we have found that the organohydrogels with the same dispersion binary solvent showed similar antifreezing and antidehydration properties, but different mechanical properties. There are two opposite factors that influence the mechanical properties of the organohydrogels for each synthesis strategy. The enhanced physical interactions between the solvent molecules (glycerol or EG) and the polymers prefer to strengthen the gels, while the solvent effect for in-situ gelling or the swelling for solvent displacement tends to weaken the gels. The competition between these two opposite factors determines the mechanical properties of the as-prepared organohydrogels. Generally, for the chemically crosslinked PAAm organohydrogels, the solvent effect or swelling-induced weakening dominates the mechanical performance of the organohydrogels, while the strengthening effect from the increased physical interactions plays a secondary role. However, for the physical gelatin organohydrogels, the glycerol induced strengthening effect plays a more important role in mechanical performance. Furthermore, compared to the in-situ gelling strategy, solvent displacement is a more universal strategy, which could synthesize the organohydrogels with the organic–water binary solvent in any proportion.

Organohydrogels with distinct antifreezing and antidehydration properties have been becoming an emerging research area. This work has studied and compared the two popular strategies for preparing the organohydrogels, and preliminarily elucidated the advantages and disadvantages of each synthesis strategies. We hope this study may provide a better understanding of the synthesis strategies of organohydrogels and provide a valuable guide to choose the suitable synthesis strategy for each organohydrogel with its application.

## Figures and Tables

**Figure 1 polymers-12-02670-f001:**
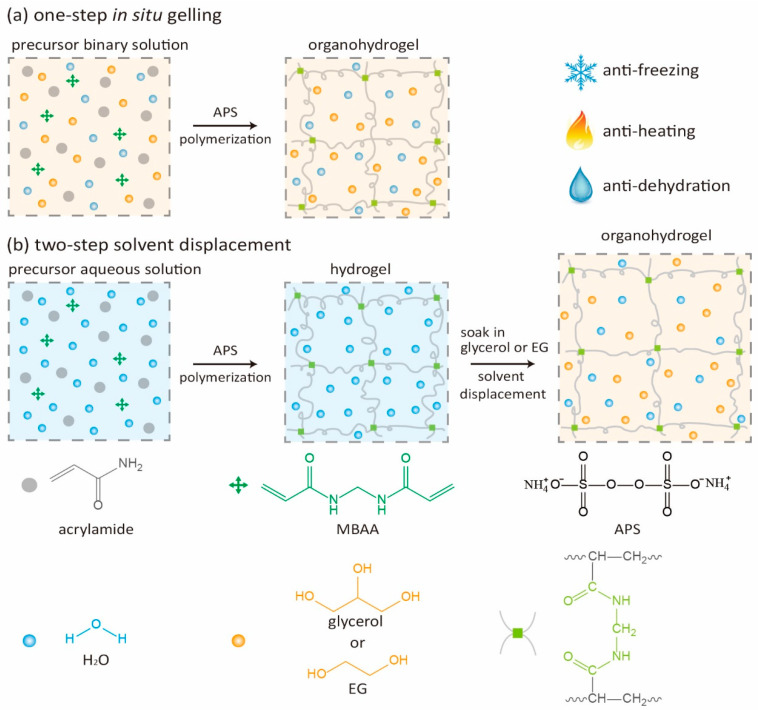
The schematic illustration for synthesizing the polyacrylamide (PAAm) organohydrogels with (**a**) one-step in-situ gelling strategy and (**b**) two-step solvent displacement strategy.

**Figure 2 polymers-12-02670-f002:**
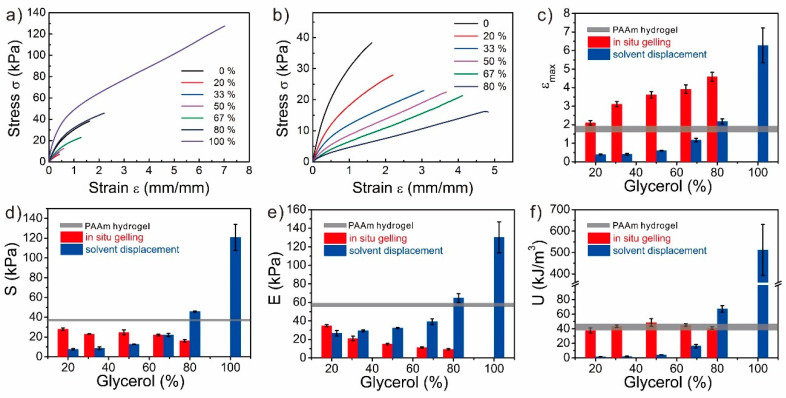
Mechanical properties of the PAAm gels with various glycerol contents from both synthesis strategies. (**a**,**b**) Stress-strain curves of the PAAm gels with various glycerol contents from (**a**) in-situ gelling and (**b**) solvent displacement. (**c**) Tensile strain, (**d**) tensile stress, (**e**) elastic modulus, and (**f**) toughness of the PAAm gels as the functions of the glycerol content in PAAm gels. The horizontal grey lines (**c**–**f**) indicate the data from PAAm hydrogels, and the width of the line represents the error range.

**Figure 3 polymers-12-02670-f003:**
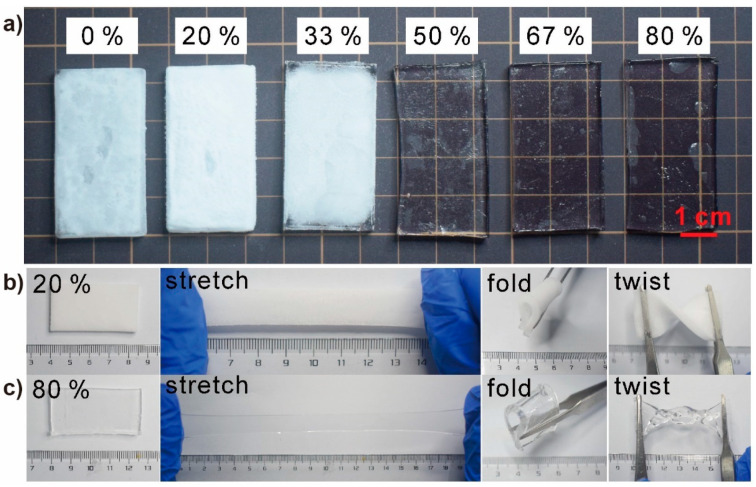
The antifreezing investigation of the PAAm gels. (**a**) Digital photograph of the PAAm hydrogel and PAAm organohydrogels from in-situ gelling after stored at −40 °C for 24 h. (**b**) The PAAm organohydrogel with 20% glycerol became opaque but maintained mechanical flexibility. (**c**) PAAm organohydrogel with 80% glycerol maintained transparency and mechanical flexibility.

**Figure 4 polymers-12-02670-f004:**
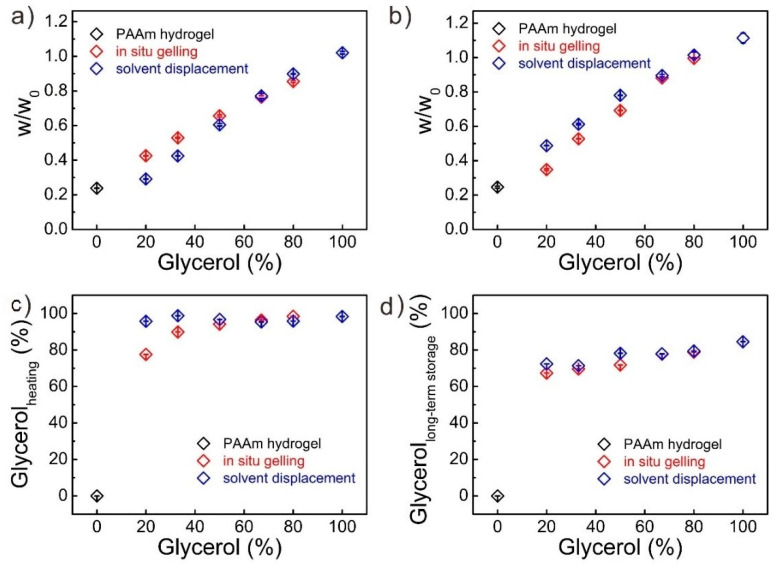
The antiheating and antidehydration investigations of the PAAm gels from both strategies. (**a**,**b**) The remaining weight ratios of the PAAm gels with different glycerol contents after (**a**) heating in the 60 °C oven for 24 h and (**b**) storing at room temperature with low humidity (25 °C and 50% humidity) for 15 days. (**c**,**d**) The calculated glycerol content in the remaining PAAm gels after (**c**) heating and (**d**) long-term storing treatments.

**Figure 5 polymers-12-02670-f005:**
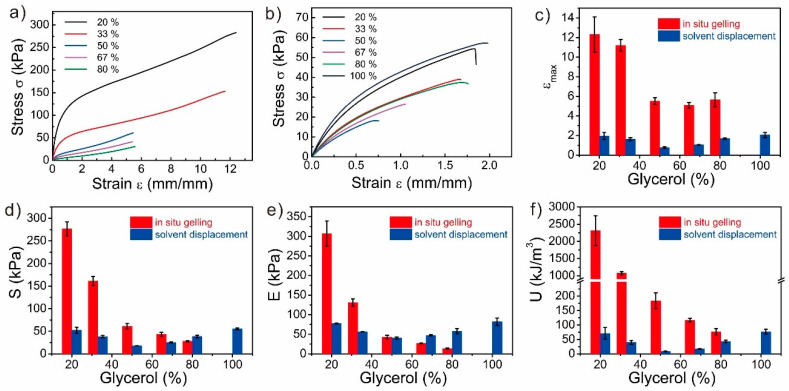
Mechanical properties of the PAAm gels with various glycerol contents after heating at 60 °C oven for 24 h. (**a**,**b**) Stress-strain curves of the PAAm gels after heating from (**a**) in-situ gelling and (**b**) solvent displacement. (**c**) Tensile strain, (**d**) tensile stress, (**e**) elastic modulus, and (**f**) toughness of the PAAm gels after heating as the functions of the glycerol content.

**Figure 6 polymers-12-02670-f006:**
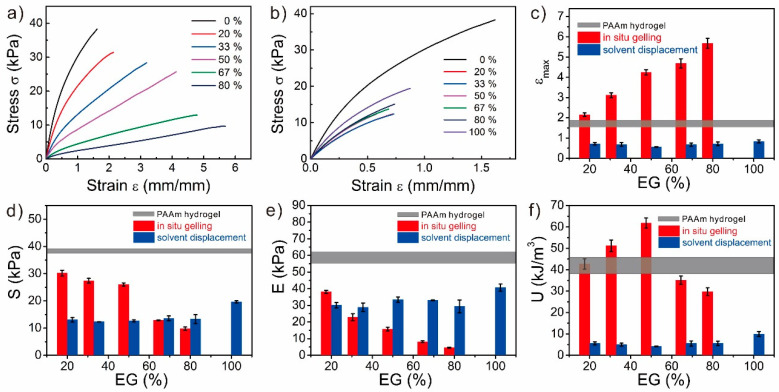
Mechanical properties of the PAAm gels with various EG contents from both synthesis strategies. (**a**,**b**) Stress-strain curves of the PAAm gels with various EG contents from (**a**) in-situ gelling and (**b**) solvent displacement. (**c**) Tensile strain, (**d**) tensile stress, (**e**) elastic modulus, and (**f**) toughness of the PAAm gels as the functions of the EG content. The horizontal grey lines (**c**–**f**) indicate the data from PAAm hydrogels, and the width of the line represents the error range.

**Figure 7 polymers-12-02670-f007:**
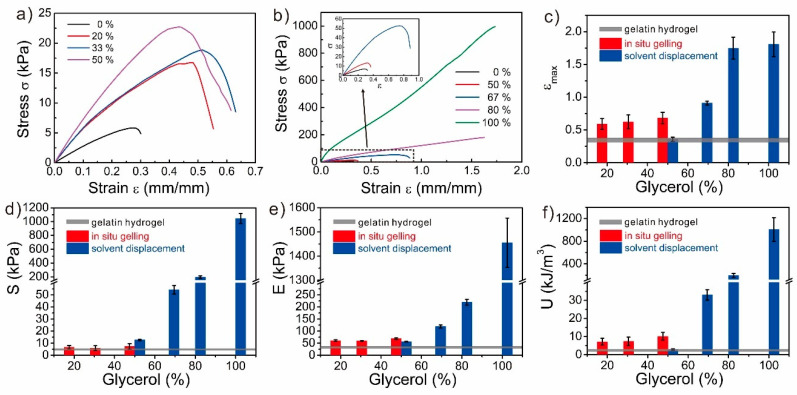
Mechanical properties of the gelatin gels with various glycerol contents from both synthesis strategies. (**a**,**b**) Stress-strain curves of the PAAm gels with various glycerol contents from (**a**) in-situ gelling and (**b**) solvent displacement. (**c**) Tensile strain, (**d**) tensile stress, (**e**) elastic modulus, and (**f**) toughness of the gelatin gels as the functions of the glycerol content. The horizontal grey lines (**c**–**f**) indicate the data from PAAm hydrogels, and the width of the line represents the error range.

**Table 1 polymers-12-02670-t001:** PAAm organohydrogels with different solvent contents.

Organic Solvent (%)	AAm(g)	H_2_O (mL)	Organic Solvent (mL)	MBAA (mg)	APS (mg)
0	7.2	30	0	21.6	360
20		24	6		
33		20	10		
50		15	15		
67		10	20		
80		6	24		

**Table 2 polymers-12-02670-t002:** Gelatin organohydrogels with different solvent contents.

Glycerol (%)	Gelatin (g)	H_2_O (mL)	Glycerol (mL)
0	5.29	30	0
20		24	6
33		20	10
50		15	15

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
