# Peer review of "Synthesis Antifreezing and Antidehydration Organohydrogels: One-Step In-Situ Gelling versus Two-Step Solvent Displacement"

_polymers, 2020, doi:10.3390/polym12112670_

Round 1

Reviewer 1 Report

This manuscript deals with organohydrogels based on PAAm and gelatin prepared by two different syntheses; swelling, mechanical and environmental tolerant properties are studied. I appreciate quite detailed analysis of obtained data which helps to clarify the both preparation strategies. Nevertheless, there are several objections and comments as suggested below which should be taken account.

  1. Surprisingly, data from swelling experiments in water (equilibrium swelling ratio) are missing. Authors refer to “ the large swelling ratio“ (ln. 192) but data are not shown. On the othe hand, the mechanical experiments were realized on hydrogels swollen in water.
  2. Swelling (determination of swelling ratios by weight and by volume) and DSC experiments are not described in Section 2.
  3. Section 3.2, Figure 2: The effect of preparation parameters (crosslinking density, polymer concentration) on mechanical properties of organohydrogels prepared on-situ should be considered. As shown in Figure 2, the PAAm organohydrogel became softer with increasing content of glycerol. At the same time as it is mentioned in lns.154-159, monomers (and probably also crosslinker) show lower solubility during in-situ synthesis as content of glycerol is increasing. The in-situ prepared organohydrogels thus could have various crosslinking density and polymer concentration despite they were prepared at constant initial conditions.
  4. There are several unclearly and unintelligibly formulated paragraphs in the text:

ln 131: For stress-strain tensile tests. The hydrogel samples were stretched until the fracture …

ln 136: Measured the weight of the organohydrogels (w0) and placed them in the 60 ºC hot oven …

ln. 272: The remaining weight ratios (a) and of the PAAm gels with different glycerol contents …

ln 274: The calculated glycerol content in the remained PAAm gels …

ln. 290: … lower mechanical properties.

5. All grahs are too small which make them illegible, mainly it is difficult to distinguish colours.

Mistakes:

ln. 68: Reference [34] is not numbered in order of appearance in the text.

Table 1: H2O

ln. 262: … the organohydrogels form in situ …

Reviewer 2 Report

Authors used two synthetic strategies, in situ gelling and the solvent displacement strategies, to prepare the anti-freezing and anti-dehydration organohydrogels, They tried to elucidate the comparative advantages of the two methods by comparing the mechanical and environmental tolerant properties of the organohydrogels prepared from both strategies. Authors used glycerol-based and ethylene glycol-based chemical polyacrylamide (PAAm) organohydrogel and the glycerol-based physical gelatin organohydrogel for this comparative study, Authors showed the similar anti-freezing and anti-dehydration properties but different mechanical properties where the mechanical properties were influenced by the enhanced physical interactions induced strengthening and solvent effect or swelling induced weakening.

I think this manscript could be accepted after the revision of the following questions.

1) Authors need to make a new schematic illustration based on the chemical structures for this research. Figure 1 did not suggest the proper scheme for this research at all.

2) Investigation of the difference of rheological properties of the gels would be necessary using oscillation angular frequency sweep test, stress strain amplitude sweep test and temperature ramp test.

3) Format of all Figures should be changed to make the conclusion more clearly, The labels should be clarified. In many case, the labels were overlapped. And more discussions for those results for all Figures are necessary based on the chemistry.

4) The red or blue X in all Figures should not be drawn. You do not have to draw the red or blue X in the X-axis in your Figures.

5) English corrections are necessary.

Round 2

Reviewer 1 Report

The manuscript has been improved and my comments were considered. I thus recommend to publish the manuscript in Polymers.

Reviewer 2 Report

I think authors made proper responses for all questions.

So, I think the manuscript can be accepted in present form.